# Genetic Variation in CCL18 Gene Influences CCL18 Expression and Correlates with Survival in Idiopathic Pulmonary Fibrosis: Part A

**DOI:** 10.3390/jcm9061940

**Published:** 2020-06-21

**Authors:** Ivo A. Wiertz, Sofia A. Moll, Benjamin Seeliger, Nicole P. Barlo, Joanne J. van der Vis, Nicoline M. Korthagen, Ger T. Rijkers, Henk J.T. Ruven, Jan C. Grutters, Antje Prasse, Coline H.M. van Moorsel

**Affiliations:** 1Centre for Interstitial Lung Diseases, Department of Pulmonology, St. Antonius Hospital, 3435 CM Nieuwegein, The Netherlands; i.wiertz@antoniusziekenhuis.nl (I.A.W.); s.moll@antoniusziekenhuis.nl (S.A.M.); n.barlo@antoniusziekenhuis.nl (N.P.B.); n.korthagen@antoniusziekenhuis.nl (N.M.K.); j.grutters@antoniusziekenhuis.nl (J.C.G.); 2Department of Respiratory Medicine, Hannover Medical School and Biomedical Research in End-stage and Obstructive Lung Disease Hannover, German Lung Research Center (DZL), 30265 Hannover, Germany; seeliger.benjamin@mh-hannover.de; 3Department of Medical Microbiology and Immunology, St. Antonius Hospital, 3435 CM Nieuwegein, The Netherlands; a.vandervis@antoniusziekenhuis.nl (J.J.v.d.V.); g.rijkers@antoniusziekenhuis.nl (G.T.R.); 4Department of Clinical Chemistry, St. Antonius Hospital, 3435 CM Nieuwegein, The Netherlands; h.ruven@antoniusziekenhuis.nl; 5Division Heart & Lungs, University Medical Center Utrecht, 3584 CX Utrecht, The Netherlands; 6Fraunhofer Institute for Toxicology and Experimental Medicine, 30625 Hannover, Germany

**Keywords:** idiopathic pulmonary fibrosis, chemokine, CCL18, single nucleotide polymorphism, survival

## Abstract

Idiopathic pulmonary fibrosis (IPF) is a progressive fibrotic disease, characterized by fibroblast proliferation and extracellular matrix deposition. CC-chemokine ligand 18 (CCL18) upregulates the production of collagen by lung fibroblasts and is associated with mortality. This study was designed to evaluate the influence of single nucleotide polymorphisms (SNPs) in the *CCL18* gene on CCL18 expression and survival in IPF. Serum CCL18 levels and four SNPs in the *CCL18* gene were analyzed in 77 Dutch IPF patients and 349 healthy controls (HCs). *CCL18* mRNA expression was analyzed in peripheral blood mononuclear cells (PBMCs) from 18 healthy subjects. Survival analysis was conducted, dependent on CCL18-levels and -genotypes and validated in two German IPF cohorts (Part B). IPF patients demonstrated significantly higher serum CCL18 levels than the healthy controls (*p* < 0.001). Both in IPF patients and HCs, serum CCL18 levels were influenced by *rs2015086* C > T genotype, with the highest CCL18-levels with the presence of the C-allele. Constitutive *CCL18* mRNA-expression in PBMCs was significantly increased with the C-allele and correlated with serum CCL18-levels. In IPF, high serum levels correlated with decreased survival (*p* = 0.02). Survival was worse with the CT-genotype compared to the TT genotype (*p* = 0.01). Concluding, genetic variability in the *CCL18*-gene accounts for differences in *CCL18* mRNA-expression and serum-levels and influences survival in IPF.

## 1. Introduction

Idiopathic pulmonary fibrosis (IPF) is a progressive fibrotic disease of the lung parenchyma, characterized by fibroblast proliferation and extracellular matrix deposition [1]. New therapeutic agents, such as nintedanib and pirfenidone, demonstrated a deceleration of disease progression [2,3], but overall survival in IPF remains drastically impaired despite these agents [4]. There is substantial inter-individual difference in the clinical course of the disease, ranging from rapid decline to periods of relative stability for many years [1,5]. To predict the disease course, an increasing number of studies investigated the use of several biomarkers in IPF, including data from multicentric randomized trials [6,7,8,9,10,11,12].

CC-chemokine ligand 18 (CCL18) is the biomarker most consistently associated with outcomes in IPF and has been studied among several interstitial lung diseases [13,14,15,16,17,18,19]. A clear relationship has been demonstrated between the elevated serum levels of CCL18 and clinical outcomes in IPF patients, including survival [16] and acute exacerbation rate [20], and has been confirmed in data from two randomized controlled trials [13]. Apart from IPF, elevated serum CCL18 levels reflected pulmonary fibrosis activity [21] and correlated with death or the progression of pulmonary disease [15,17,22] in patients with systemic sclerosis (SSc)-associated ILD.

CCL18 is predominantly expressed by alveolar macrophages and occurs at relatively high levels in lung tissue [23]. In response to CCL18, lung fibroblasts from healthy adults showed an increased expression of collagen mRNA [24]. Furthermore, it was demonstrated that alveolar macrophages from patients with pulmonary fibrosis show an alternatively activated phenotype, which up-regulates the production of collagen by lung fibroblasts through the production of CCL18 [14]. As fibroblast contact and exposure to collagen increases spontaneous CCL18 production by alveolar macrophages, a positive feedback loop was suggested that maintains fibrosis.

The gene encoding CCL18 is small, positioned at the q arm of chromosome 17 and consist of three exons with a number of single nucleotide polymorphisms (SNPs), including the SNPs *rs2015086* and *rs712040*, present in the region. In the macrophages of subjects with the A > G genotype of the *rs2015086* SNP, a threefold higher gene expression was found compared to those with the A/A genotype [25,26]. We hypothesized that genetic variation in the *CCL18* gene might be associated with increased CCL18 expression and may predispose to an unfavorable prognosis in subjects with IPF.

## 2. Experimental Section

### 2.1. Patients and Clinical Data

A retrospective cohort study was performed in an IPF cohort from the Netherlands. IPF patients diagnosed at St. Antonius Interstitial Lung disease Center of Excellence, Nieuwegein, The Netherlands between 1998 and 2007 were assigned to the derivation cohort (Part A, presented herein). Diagnoses were reviewed and patients were included if the 2011 ATS/ERS criteria were met [27]. Medical records were retrieved to determine survival status, cause of death and baseline characteristics, including age, sex, percentage of predicted forced vital capacity (%FVC) and diffusion capacity for carbon monoxide (%DLCO) were recorded. The findings were then validated in two independent prospectively-recruited German IPF cohorts (presented in Part B of this work). The study protocol was approved by the local Ethical Committee of the St. Antonius Hospital (registration number R05-08A) and all of the subjects gave written informed consent.

### 2.2. Blood Sampling

Serum and blood for DNA extraction were collected at diagnosis. All of the serum samples of patients with the CC and CT genotype were analyzed and, additionally, a random sample from patients with TT were analyzed. For details on DNA analyses, we refer to the supplement.

### 2.3. Serum CCL18 Measurement by Monoplex Bead Array

CCL18 levels were analyzed using a monoplex suspension bead array system in the derivation cohort. In the validation cohort an ELISA test was used to determine CCL18 levels, as described earlier in other articles [16]. For further details on CCL18 analysis, we refer to the supplement.

### 2.4. Single Nucleotide Polymorphism (SNP) Genotyping and mRNA Expression Analysis

Patients were genotyped for multiple SNPs. Two SNPs, with presumed functionality in the promotor region [26], were genotyped (rs712040, rs2015086). In addition, two haplotype tagging SNPs (rs712042, rs712044) were selected to cover genetic variability in the CCL18 gene. The expressions of CCL18 mRNA in peripheral blood mononuclear cells (PBMCs) from 18 healthy donors were analyzed by quantitative RT-PCR amplification. Please refer to the Appendix A for further details on genotyping and mRNA expression analysis.

### 2.5. Statistical Analysis

Genotypes were tested for Hardy–Weinberg equilibrium using the website (https://ihg.gsf.de/cgi-bin/hw/hwa1.pl). Linkage disequilibrium (r2) was calculated using the computer program Haploview 4.1, Broad Institute at Massachusetts Institute of Technology at Harvard University, MA, USA [28]. Haplotypes were determined using Phase v2.1 (Department of Human Genetics, University of Chicago, IL, USA) [29]. The data are presented as medians and interquartile ranges (IQR). Statistical comparisons were made with the use of the Mann–Whitney U test for two groups or Kruskal–Wallis for more than two groups. Receiver operating curves (ROC) were determined to define the optimal cut-off point for distinct serum CCL18 concentrations with the highest predictive accuracy for one year survival. In addition, the Kaplan–Meier method was used for survival analyses. For the analysis of correlation, log-transformation was used to reach near normal distribution. The correlation between mRNA expression and serum levels was assessed using Pearson’s correlation coefficients. Statistical analysis was performed using IBM SPSS statistics software for Windows (version 22.0; IBM, Armonk, NY, USA) and Graphpad PRISM 5 (GraphPad Software, La Jolla, CA, USA) and RStudio version 1.2.5033 (RStudio Inc, Boston, MA, USA). Statistical significance was considered at a value of *p* < 0.05.

## 3. Results

### 3.1. Derivation Cohort

A total of 77 IPF patients were included in this study: 58 male, 19 female, median age 61.4 years [IQR 54.1–71.6]). No significant differences were demonstrated in the baseline characteristics between carriers of the different genotypes in the derivation cohort, as shown in Table 1. All of the patients were derived from a pre-antifibrotic treatment cohort.

### 3.2. CCL18 Genotypes and Allele Carrier Frequencies

DNA was present for 77 patients in the derivation cohort and for 349 healthy subjects (139 male (40%), 210 female (40%), median age 39.4 years, [IQR 28.3–49.1]). All of the controls and 71 of 77 IPF patients (93%) were of European descent. Five IPF patients (6%) were of North American descent and one (1%) of North African descent. Differences in ethnical background were not statistically different between the healthy controls and IPF patients (*p* = 0.184).

The genotypes and allele carrier frequencies in the 77 IPF patients of the derivation cohort and 349 controls are summarized in Table 2. The healthy controls and IPF patients were in Hardy–Weinberg equilibrium for all polymorphisms. A comparison of the SNPs in the *CCL18* gene revealed no significant differences in allele frequencies between the IPF patients and controls. The following SNPs, including the two SNPs with presumed functionality, showed strong linkage disequilibrium (LD): *rs712042*, *rs2015086* and *rs712040*; 76 < r2 < 0.90, as shown in Figure 1. Additionally, based on four SNPs, only three haplotypes were constructed with a frequency > 5%. Haplotype frequencies were not significantly different between the IPF patients and healthy controls. Although *rs712042* showed the strongest LD, a presumably functional SNP was preferred for further evaluation. Therefore, the *rs2015068* polymorphism was selected for analysis.

### 3.3. Serum CCL18 Levels and CCL18 Genotypes

Serum at the time of diagnosis was available from 61 of 77 patients in the derivation cohort (79%). The characteristics of patients with available serum did not differ from the total group, as shown in Appendix A. The serum levels were further measured in a selection of 204 healthy controls (86 male (42%), 118 female (58%), median age 40.1 years [IQR 29.5 – 51.0]) who were enriched for the presence of the minor allele rs2015086. The characteristics of these healthy controls did not differ from the total group of controls (p ranging 0.768–0.999). Analysis of the derivation cohort showed that serum CCL18 levels were significantly higher in the IPF patients (645 ng/mL [IQR 393 – 847]) compared with the healthy controls (185 ng/mL [IQR 123-272]), *p* < 0.0001, as shown in Figure 2A. The serum CCL18 levels (median 642 ng/mL [IQR 429-943] in eight patients who received low dose corticosteroids were not significantly different from those of the IPF patients who did not receive immunosuppressive therapy (median 652 ng/mL; IQR 400-856; *p* = 0.880) at the time of sampling. 

In the healthy controls, significant differences in serum CCL18 levels were observed between the carriers and non-carriers of the C-allele of the rs2015086 polymorphism; TT 151 ng/mL (IQR 109-224), CT / CC 239 ng/mL (IQR 152-328), (*p* < 0.0001), as shown in Figure 2B. No significant differences were found in CCL18 levels between the CT-group (241 ng/mL; IQR 156-327) and CC-group (212 ng/mL; IQR 137-483; *p* = 0.834).

Pronounced differences in the serum CCL18 levels were observed between genotypes of the rs2015086 polymorphism in the IPF patients of the derivation cohort; TT 585 ng/mL (IQR 340 –793) and CT 817 ng/mL (IQR 681 – 1278), *p* = 0.002, as shown in Figure 2C. No differences were present in the baseline characteristics between the patients with CT and TT genotypes of rs2015086, as shown in Table 1.

### 3.4. CCL18 Genotypes and mRNA Expression

The expression of *CCL18* mRNA in PBMCs was analyzed in 18 healthy controls. Six subjects had genotype CT for the *rs2015086* SNP and 12 subjects had TT. The subjects with the CT genotype had a four-fold higher gene expression (3.0 × 10^−5^; [IQR 1.8 × 10^−5^ − 7.7 × 10^−5^]) than the subjects with TT (7.4 × 10^−6^ [IQR 1.1 × 10^−6^ −1.8 × 10^−5^], *p* = 0.007), as shown in Figure 3A. *CCL18* mRNA expression correlated significantly with serum CCL18 levels (*r* = 0.73, *p* = 0.002), as shown in Figure 3B. Analysis of CCL18 mRNA expression for the *rs712040, rs712042* and *rs712044* SNPs, respectively, showed similar, but non-significant higher gene expression in the CT-genotypes compared with the TT-genotypes. Weak to moderate, non-significant correlations between the *CCL18* mRNA and serum CCL18 levels were found as well for the *rs712040, rs712042* and *rs712044* SNPs, respectively.

### 3.5. Progression of Disease and Survival in IPF Patients

Progression of disease was evaluated for the IPF patients, based on FVC change and/or the clinical suspicion of an acute exacerbation. Of the 77 patients, a total of nine patients (13.2%) had a clinical suspicion of an acute exacerbation. A FVC decline of more than 10% after one year was considered as FVC progression. The FVC values at one-year follow-up were available for 41 out of 77 IPF patients. The median FVC change after one year was −4.5% (IQR −9.4–2.3). Altogether, 18 patients (44%) showed a progression of disease. A trend towards higher baseline CCL18 levels (652 ng/mL; IQR 505-791) was observed in subjects with a progression of disease compared with those without progression (592.8 ng/mL; IQR 328-853; *p* = 0.699). No significant differences were found for the progression of disease between patients with CT and TT genotypes of *rs2015086* (*p* = 0.342). The frequencies of acute exacerbations did not differ between patients with the CT-genotype (14.3%) and TT-genotype of *rs2015086* (13.0%; *p* = 0.896).

The median survival in the derivation cohort was 35 months (95% CI 21.1–48.7). Within the study period, 50 out of 77 IPF patients died (65%), and one patient was lost to follow-up. ROC analyses showed that the highest area under the curve (AUC) was calculated for a serum CCL18 concentration of 500 ng/mL (AuC = 0.72). According to this cut-off level, patients were categorized as having high (serum CCL18 > 500 ng/mL) or low levels (serum CCL18 < 500 ng/mL). The median survival in the CCL18^low^-group was 50.4 months (95% CI 31.9–68.9) and differed significantly from that of 27.6 months (95% CI 8.1–47.0) in the CCL18^high^-group (*p* = 0.02), as shown in Figure 4A.

Survival was also analyzed for dependency on the CCL18 genotype. Patients with the *rs2015086* CT genotype showed a significantly worse survival (median 14.3 months (95% CI 0.0–35.9)) compared to the TT genotype (median 37.2 months (15.4–58.9), *p* = 0.01), as shown in Figure 4B. Patients were censored from the survival analysis if they were alive at end of the follow-up (*n* = 15) or had received lung transplantation (*n* = 11). Censored patients were genotyped CT (*n* = 4) and TT (*n* = 22).

The survival rates were also analyzed in three groups based on a combination of the serum CCL18 level and genotype: CCL18^low/TT^ group, median survival 50.4 months (95% CI 25.4–75.4); CCL18^high/TT^ group, median survival 37.2 months (95% CI 13.1–61.3); CCL18^high/CT^ group, median survival 14.3 months (95% CI 1.4–27.2) (*p* = 0.03), as shown in Figure 4C. The p-value was calculated via the Log Rank Test using Kaplan–Meier curves. There were no patients with low CCL18 and genotype CT.

## 4. Discussion

CCL18 protein levels are known as a promising biomarker for IPF and this study showed that the results are dependent on the *CCL18* genotype. The *rs2015086* CT polymorphism in the *CCL18* gene contributes to inter-individual differences in healthy controls, with individuals carrying the C-allele having the highest CCL18 mRNA and protein expression. A similar genotypic effect on serum CCL18 levels was observed in patients with IPF, even though the mean serum levels showed a 3.5-fold increase compared to the healthy controls. Both the elevated serum CCL18 levels and CT genotypes were related to a significantly diminished long-term survival in IPF. Patients with the worst survival rate on the basis of high serum CCL18 levels could be subdivided into intermediate and worse survival rates according to genotype.

Serum CCL18 concentrations reflect pulmonary fibrotic activity in patients with idiopathic interstitial pneumonias (IIPs) and systemic sclerosis with pulmonary involvement [15,21]. Prasse et al. demonstrated that increased serum CCL18 levels were associated with increased short-term (24 months follow-up) mortality in IPF patients [16]. In our study, we independently confirmed these results and added to this finding the predictive value of serum CCL18 for long-term survival. Further, we showed that serum CCL18 levels were genotype dependent. Subjects with the CT genotype displayed higher constitutive serum CCL18 levels. The CT-genotype of *rs2015086* caused a four-fold higher mRNA expression in PBMCs from healthy controls. The influence of genotypes on mRNA expression in IPF has not been investigated. Interestingly, Hägg et al. described that patients with carotic artery plaques and the CT genotype of *rs2015086* had a three-fold higher gene expression level in macrophages than subjects with the TT genotype [25]. This is in the same order of magnitude as our results and, with that, both the genotype–mRNA correlation and the protein–survival correlation have been demonstrated twice independently.

At presentation, patients with the *rs2015086* CT genotype did not show any significant differences in demographics or lung function parameters compared to patients with the TT genotype, as shown in Table 1. We found an association between the *rs2015086* polymorphism and serum CCL18 levels in both the controls and patients. Besides that, one may question whether higher constitutive CCL18 levels predispose to fibrotic disease. In order to investigate whether the carriage of the C-allele predisposes to IPF, we compared allele frequencies between the cases and controls and found no significant differences, even though we had 80% power to detect an OR ≥ 2.1 under a dominant gene model. The absence of an association shows that the carriage of the C-allele does not significantly predispose to IPF, however, due to limited sample size, small predisposing effects may still exist.

Alveolar macrophages are the main source of CCL18 in the lungs and show an alternatively activated phenotype in IPF [14]. Fibroblast contact and exposure to collagen increases CCL18 production by alveolar macrophages, and these macrophages up-regulate collagen production by lung fibroblasts via the production of CCL18. As such, we hypothesize that an increase in CCL18 does not precede disease but occurs during disease, and that the degree to which CCL18 increases is dependent on the presence of the C-allele.

In the search for a biomarker to predict prognosis in IPF, a great number of studies have focused on proteins in serum and BALF. This study is the first to show a genetic polymorphism correlating with serum biomarker levels and with disease course in IPF patients. Genotyping IPF patients for the *rs2015086* SNP in the CCL18 gene may, therefore, add substantial information to the interpretation of serum CCL18 levels with regard to the prediction of the disease course.

This study has some limitations related to the retrospective nature and relatively small sample size of the IPF cohort. Furthermore, the inclusion period started before 2011 and thereby, patients were selected before the era of new antifibrotic drugs, such as pirfenidone and nintedanib. We could not determine the potential negative effects of immunosuppressive therapy. The difference in age and sex between the IPF patients and healthy controls may have influenced the results. However, with this approach we were able to evaluate a long follow-up period to determine the long-term effect of the biomarker. To address these limitations, in Part B of this work we prospectively validated these findings in two independent German cohorts, one pre-antifibrotic and one with the majority of patients receiving anti-fibrotic therapy.

As serum CCL18 levels increase in IPF and influence the disease course, it can be hypothesized that the *rs2015086* polymorphism may show similar effects in other fibrotic lung diseases. CCL18 expression is increased in patients with systemic sclerosis and in hypersensitivity pneumonitis [13,17,21,30]. Morbidity and mortality in these diseases are mainly caused by pulmonary fibrosis. Both diseases show a subset of patients who develop a phenotype in which progressive pulmonary fibrosis is the major cause of death. Further research is needed to investigate whether genetic variation in the CCL18 gene influences serum levels and disease course in systemic sclerosis and hypersensitivity pneumonitis.

Carriers of the CT genotype are at a disadvantage in terms of higher CCL18 levels and diminished prognosis. Interrupting the positive feedback loop by blocking CCL18 might be an interesting therapeutic intervention. IPF is a relentlessly progressive disease and therapy is not curative and, in general, does not stabilize the disease [3,31,32]. As increased CCL18 levels stimulate fibroblasts to produce collagen, inhibiting CCL18 activity may directly inhibit fibrogenesis. Patients with the CT genotype may especially benefit from a CCL18 blockade as they have the highest serum CCL18 levels.

## 5. Conclusions

In conclusion, we showed that genetic variability in the *CCL18* gene accounts for significant differences in *CCL18* mRNA expression and serum levels, and was shown to have a modifying role over the course of IPF. Our findings emphasize the value of serum CCL18 as a prognostic marker for IPF. Moreover, we confirmed in a replication cohort that future studies concerning CCL18 should take into account that mRNA and protein expression are influenced by genetic polymorphisms in the *CCL18* gene. The findings of this study are validated in Part B of this work.

## Figures and Tables

**Figure 1 jcm-09-01940-f001:**
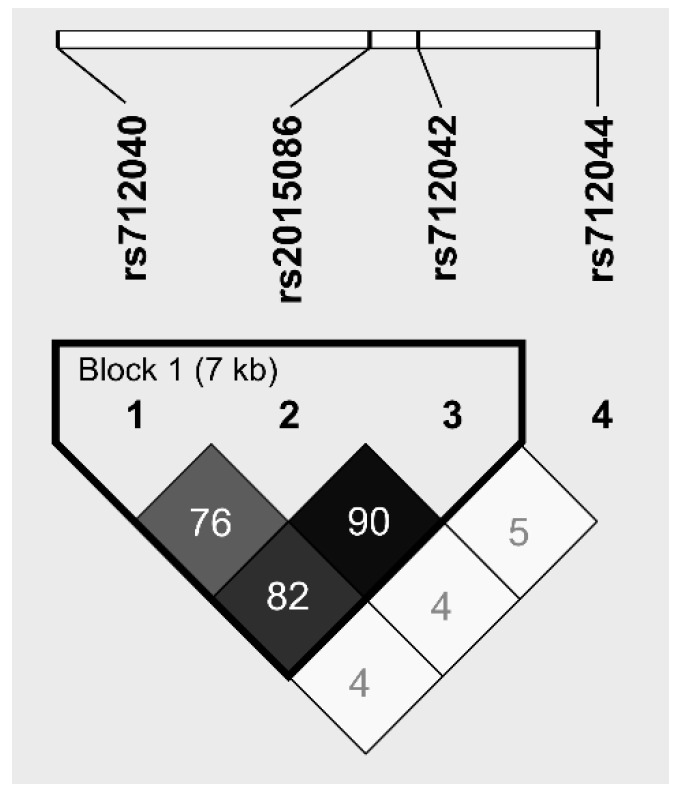
Linkage Disequilibrium (LD) map of four SNPs in the CCL18 gene: *rs712040* (location: CCL18 promotor), *rs2015086* (location: CCL18 promotor), *rs712042* (location: CCL18 intron), *rs712044* (location: CCL18 intron). The dark squares represent high r2 values and the triangle represents a haplotype block.

**Figure 2 jcm-09-01940-f002:**
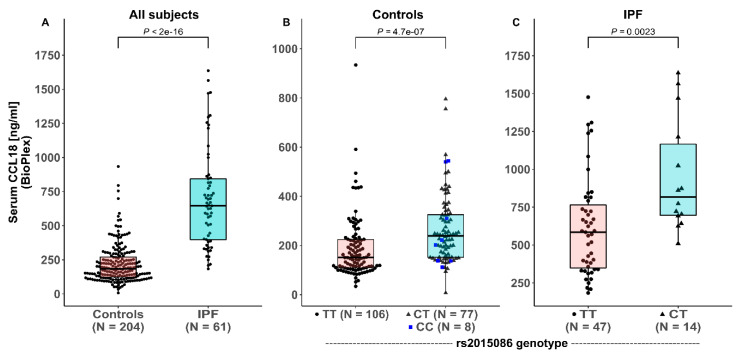
Serum CCL18 levels in healthy controls and patients with idiopathic pulmonary fibrosis (IPF) in the derivation cohort (**A**), depending on *rs2015086* genotype in healthy controls (**B**), and IPF patients (**C**).

**Figure 3 jcm-09-01940-f003:**
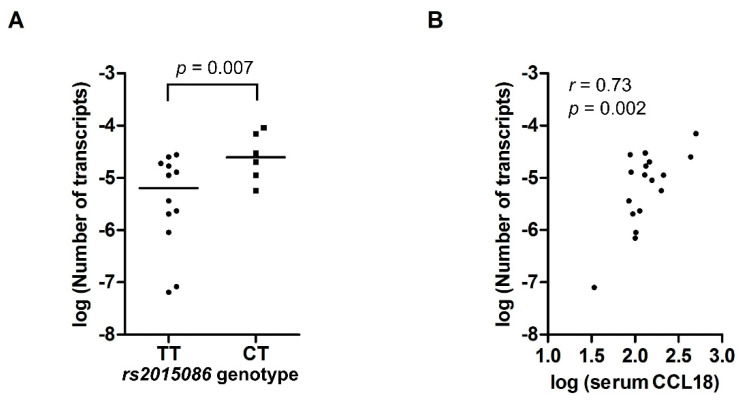
(**A**) mRNA expression of *CCL18* in PBMCs from 18 healthy controls, expressed as the number of *CCL18* transcripts per copy of β-actin, according to *rs2015086* genotype. (**B**) Scatterplot showing the correlation between serum CCL18 levels and the number of *CCL18* mRNA transcripts per copy of β-actin. Values on the X and Y-axis represent log-transformed values.

**Figure 4 jcm-09-01940-f004:**
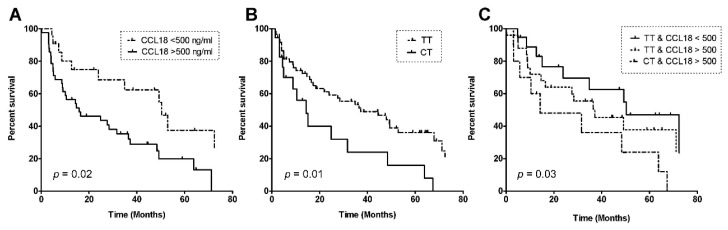
Kaplan–Meier curves for survival in patients with idiopathic pulmonary fibrosis, depending on a serum CCL18 level cut-off of 500 ng/mL (**A**); *rs2015086*-genotype (**B**), and a combination of CCL18 cut-off and *rs2015086*-genotype (**C**). P-values were calculated via the Log Rank Test using Kaplan–Meier curves.

**Table 1 jcm-09-01940-t001:** Baseline characteristics in IPF patients divided by *CCL18 rs2015086* genotype in the derivation cohort.

Characteristics	Derivation Cohort (*n* = 77)			
*rs2015086* Genotype	All (*n* = 77)	CT (*n* = 18)	TT (*n* = 59)	*p*
Male, *n* (%)	58 (75)	11 (61)	47 (79)	0.200
Age, years (IQR)	61.4 (54–72)	61.3 (53–75)	62.8 (52–74)	0.900
Former or active smoker, *n* (%)	59 (76)	11 (61)	48 (81)	0.950
Baseline %FVC predicted (IQR)	75.7 (62–87)	73.7 (52–88)	75.7 (63–89)	0.700
Baseline %DLCO predicted (IQR)	42.5 (33–56)	40.4 (32–64)	47 (31–60)	0.300

IQR (interquartile range); %FVC (Percent of predicted forced vital capacity); %DLCO (Percent of predicted diffusion capacity for carbon monoxide with single breath).

**Table 2 jcm-09-01940-t002:** Allele carrier and genotype frequencies in patients with IPF and healthy controls.

*Polymorphism*	Allele and Genotype	IPF (*n* = 77)	Healthy Controls (*n* = 349) *
rs712040	C	16 (10%)	86 (12%)
	T	138 (90%)	612 (88%)
	CC	0 (0%)	7 (2%)
	CT	16 (21%)	72 (21%)
	TT	61 (79%)	270 (77%)
rs2015086 *	C	18 (12%)	101 (15%)
	T	136 (88%)	593 (85%)
	CC	0 (0%)	10 (3%)
	CT	18 (23%)	81 (23%)
	TT	59 (77%)	256 (74%)
rs712042	A	136 (88%)	597 (86%)
	G	18 (12%)	101 (14%)
	AA	59 (77%)	258 (74%)
	AG	18 (23%)	81 (23%)
	GG	0 (0%)	10 (3%)
rs712044	A	97 (63%)	480 (69%)
	G	57 (37%)	218 (31%)
	AA	33 (43%)	172 (49%)
	AG	31 (40%)	136 (39%)
	GG	13 (17%)	41 (12%)

* healthy controls: *n* = 347, due to missing genotypes in two controls. No significant differences were observed.

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
