# Peer review of "Genetic Variation in CCL18 Gene Influences CCL18 Expression and Correlates with Survival in Idiopathic Pulmonary Fibrosis: Part A"

_jcm, 2020, doi:10.3390/jcm9061940_

Round 1
Reviewer 1 Report
The data presented in this study are clear and persuasive. As shown below, it would be a better paper if authors could supplement the data that readers want to know.
1. The authors present the prognosis of IPF depends on genotype of rs2015086 and serum CCL18 levels. There are various causes of death from IPF, including disease progression, lung cancer, pneumonia, and other diseases. However, many of the patients die with disease progression with or without acute exacerbation of IPF. I speculate that the effect of genotype or CCL18 level on survival may be the result of affecting IPF disease progression. Therefore, it is better to specify the data for delta FVC as an index of disease progression by genotype and CCL18 level. Could you present these data?
2. Acute exacerbation of IPF is usually associated with a poor prognosis. It has been reported that CCL18 production by BAL cells increased during acute exacerbations and baseline levels of CCL18 production by BAL cells were significantly predictive for the development of future acute exacerbation (PLoS One. 2015 Jan 15;10(1):e0116775). Is there difference in the frequency of acute exacerbations or the time to first acute exacerbations depending on genotype?
3. Statistical analysis may be difficult due to the small number of cases, but was there a difference in CCL18 levels between CC and CT genotype in control group? Please mention this point in the results.
4. Line 239: Isn’ t table 1 correct, not table3?
Author Response
Reviewer 1
The data presented in this study are clear and persuasive. As shown below, it would be a better paper if authors could supplement the data that readers want to know.
1. The authors present the prognosis of IPF depends on genotype of rs2015086 and serum CCL18 levels. There are various causes of death from IPF, including disease progression, lung cancer, pneumonia, and other diseases. However, many of the patients die with disease progression with or without acute exacerbation of IPF. I speculate that the effect of genotype or CCL18 level on survival may be the result of affecting IPF disease progression. Therefore, it is better to specify the data for delta FVC as an index of disease progression by genotype and CCL18 level. Could you present these data?
Response from the authors
Thank you for your suggestion. We added delta FVC values of IPF patients in our cohort, expressed as change after one year.
Next, we evaluated progression of disease in IPF patients, based on FVC change and/or the clinical suspicion of an acute exacerbation. An FVC decline of more than 10% after one year was considered as FVC progression. FVC values at one-year follow-up were availabe for 41 out of 77 IPF patients. Median FVC change after one year was -4.5% (IQR -9.4 – 2.3) and a total of 18 patients (44%) showed progression of disease. A trend towards higher baseline CCL18 levels (652 ng/ml; IQR 505-791) was observed in subjects with progression of disease compared with those without progression (592.8 ng/ml; IQR 328-853; p = 0.699). No significant differences were found in median FVC change (p =0.483), nor in progression between patients with CT and TT genotypes of rs2015086 (p = 0.342).
We added this additional analysis to the results sections (paragraph ‘Progression of disease and survival in IPF, lines 205-213’).
2. Acute exacerbation of IPF is usually associated with a poor prognosis. It has been reported that CCL18 production by BAL cells increased during acute exacerbations and baseline levels of CCL18 production by BAL cells were significantly predictive for the development of future acute exacerbation (PLoS One. 2015 Jan 15;10(1):e0116775). Is there difference in the frequency of acute exacerbations or the time to first acute exacerbations depending on genotype?
Response from the authors
We evaluated progression of disease in IPF patients, based on FVC change and/or the clinical suspicion of an acute exacerbation. An FVC decline of more than 10% after one year was considered as FVC progression. FVC values at one-year follow-up were availabe for 41 out of 77 IPF patients. Median FVC change after one year was -4.5% (IQR -9.4 – 2.3) and a total of 18 patients (44%) showed progression of disease. A trend towards higher baseline CCL18 levels (652 ng/ml; IQR 505-791) was observed in subjects with progression of disease compared with those without progression (592.8 ng/ml; IQR 328-853; p = 0.699). No significant differences were found for progression of disease between patients with CT and TT genotypes of rs2015086 (p = 0.342)
We added this additional analysis to the results section (paragraph ‘Progression of disease and survival in IPF’).
3. Statistical analysis may be difficult due to the small number of cases, but was there a difference in CCL18 levels between CC and CT genotype in control group? Please mention this point in the results.
Response from the authors
In healthy controls, significant differences in CCL18 serum levels were observed between the carriers and non-carriers of the C-allele of the rs2015086 polymorphism; TT 151 ng/ml (IQR 109-224), CT / CC 239 ng/ml (IQR 152-328), (p < 0.0001) (see Figure 2B). No significant differences were found in CCL18 levels between the CT-group (241 ng/ml; IQR 156-327) and CC-group (212 ng/ml; IQR 137-483; p = 0.834).
We added these additional result concerning differences in CCL18 levels between the CT and CC groups above to the results section (paragraph ‘Serum CCL18 levels and CCL18 genotypes’, lines 174-176).
4. Line 239: Isn’ t table 1 correct, not table3?
Response from the authors
Table 1 is indeed correct, we adjusted the text (line 265)
Reviewer 2 Report
- It is unclear why these SNPs were selected. Is there previous work which indicates them?
- No comparison can be made between IPF patients and controls as the ages are not only significantly different between the groups but also because there is no overlap is ages meaning that correction based on age cannot be done.
- Were any covariates adjusted for when assessing CCL18 expression?
- Table 1: Why is smoker information missing for the CT and TT groups?
- Where are the clinical characteristic for the controls?
- Figure 1: Annotation of the gene location on the figure would add a meaningful component.
- Although it is noted on line 156 that the subset populations did not have different characteristic than the whole group, this data should be reported in the supplement for the 66 IPF cases and 204 controls used for this analysis (there is also no note of these patients having similar characteristics to the larger population).
- Line 160-162, please provide the mean expression and SEM for the 8 patients on corticosteroids and those not on corticosteroids.
- What is the ethnic background of the patients in the population? Is this different between the cases and controls or the different genotypes?MAF for this SNP varies between 0.05 and > 0.4 based on the population and could drive some of the observations.
- Although there was high LD across 3 of the 4 SNPs genotyped, there is no confirmation of the association between CCL18 expression and those genotypes. What about the genotype that was not in LD?
- Legend for Fig 4C is too small to differentiate the different groups.
- Lines 203-206, is this a three-way analysis? Interaction? Additive model? How was the p-value calculated?
- Line 224 used the abbreviated IIPs without previously defining.
- Line 230, clarify the “C genotype”
- It is likely that you see no difference in allele frequency between your cases and controls because all of the controls are too young to have a typical form of IPF. This should be address in the discussion line 241-246
Author Response
Reviewer 2
1. It is unclear why these SNPs were selected. Is there previous work which indicates them?
Response from the authors
CCL18 levels and CCL18 gene expression were found to be increased in macrophages of subjects with atherosclerosis compared with healthy controls [Hägg DA, et al. Expression of chemokine (C-C motif) ligand 18 in human macrophages and atherosclerotic plaques. Atherosclerosis. 2009] As the authors suggested that promotor region polymorphisms affected the CCL18 gene expression, a SNP was selected within 1000 bp from translation start with the highest minor allele frequency, being rs2015086. Macrophages with the A/G genotype of the rs2015086 SNP had threefold higher gene expression compared to macrophages from subjects with the A/A genotype.
In HIV research, the genetic variation in chemokines including CCL18 was evaluated as well [Modi WS et al. Genetic variation in the CCL18-CCL3-CCL4 chemokine gene cluster influences HIV type 1 transmission and AIDS disease progression. Am J Hum Genet. 2006]. Gene positions for 21 SNPs were genotyped in the CCL18 gene, including rs2015086 (CCL18 promotor) and rs712040 (CCL18 promotor).
We added evidence about SNPs in CCL18 to the manuscript (paragraph ‘introduction’, lines 70-72)
2. No comparison can be made between IPF patients and controls as the ages are not only significantly different between the groups but also because there is no overlap is ages meaning that correction based on age cannot be done.
Response from the authors
Thank you for your remark. In general, allele frequencies are not dependent on age. Furthermore, IPF is a rare disease occurring in 494 per 100.000 persons [Raghu G et al. Idiopathic pulmonary fibrosis in US Medicare beneficiaries aged 65 years and older: Incidence, prevalence, and survival, 2001-11. Lancet Respir Med. 2014]. It is therefore unlikely that any- or a significant portion- of the controls will develop IPF in the future. However, we acknowledge that a difference in age exists between the two groups and added this observation to the discussion. We added to the discussion that the difference in age between IPF and healthy controls should be taken into account (lines 287-288).
3. Were any covariates adjusted for when assessing CCL18 expression?
Response from the authors
As described in the supplement in paragraph ‘RNA expression analysis’: during the process of RNA expression analysis, the copy number of the CCL18 was normalized by the housekeeping gene β-actin, and is presented as the number of transcripts per 1 copy of β-actin [Thellin O et al. Housekeeping genes as internal standards: Use and limits. J Biotechnol. 1999]. CCL18 expression was not adjusted for age or sex, we added this last observation as a discussion point to the discussion part (lines 287-288).
4. Table 1: Why is smoker information missing for the CT and TT groups?
Response from the authors
Thank you for this sharp observation, this information was missing by mistake. We added the missing values to table 1. There were no statistical differences in smoking-status between the CT and TT groups (p = 0.950).
5. Where are the clinical characteristic for the controls?
Response from the authors
The clinical characteristics of healthy subjects are described in the text (paragraph ‘CCL18 genotypes and allele carrier frequencies’, lines 163-166). DNA was present for 349 healthy subjects (139 male (40%) 210 female (60%), median age 39.4 years, [IQR 28.3 – 49.1]). All controls were of European descent. Allele carrier and genotype frequenties of healthy controls are listed in table 2.
6. Figure 1: Annotation of the gene location on the figure would add a meaningful component.
Response from the authors
Thank you for your suggestion, we added annotations of the located genes in the description of Figure 1 (lines 155-158).
7. Although it is noted on line 156 that the subset populations did not have different characteristic than the whole group, this data should be reported in the supplement for the 66 IPF cases and 204 controls used for this analysis (there is also no note of these patients having similar characteristics to the larger population).
Response from the authors
Thank you for your suggestion, we described the characteristics of IPF patients and healthy controls with available serum compared to all IPF patients and all healthy controls in the supplementary table 1 and the text (paragraph ‘serum CCL18 levels and CCL18 genotypes’, lines 162-166) respectively.
8. Line 160-162, please provide the mean expression and SEM for the 8 patients on corticosteroids and those not on corticosteroids.
Response from the authors
Thank you for your suggestion, we added the median serum CCL18 levels of the eight IPF patients who received low dose corticosteroids compared to non-corticosteroids IPF patients to the paragraph ‘serum CCL18 levels and CCL18 genotypes’ (currently lines 168-171). In addition, we showed the p-value (0.880) of the difference in median serum CCL18 levels between these groups.
9. What is the ethnic background of the patients in the population? Is this different between the cases and controls or the different genotypes? MAF for this SNP varies between 0.05 and > 0.4 based on the population and could drive some of the observations.
Response from the authors
Thank you for your suggestion to describe the ethnic background more detailed. All healthy controls and 71/77 IPF (93%) were of European descent. Five IPF patients (6%) were of North-American descent and one (1%) of North-African descent. Differences in ethnical background were not statistically different between healthy controls and IPF patients (p=0.184)
10. Although there was high LD across 3 of the 4 SNPs genotyped, there is no confirmation of the association between CCL18 expression and those genotypes. What about the genotype that was not in LD?
Response from the authors
We also analysed the expression of CCL18 mRNA in PBMCs in 18 healthy controls for the other SNPs, namely rs712040, rs712042 and rs712044.
Subjects with the CT genotype for rs712040 SNP had a non-significant higher gene expression (2.0 x10-5 [IQR 0.7 x10-5 – 6.0 x10-5]) compared with TT genotype (1.1x10-5 [IQR 0.2x10-5 – 2.0x10-5, p = 0.084). Analysis for SNP rs712042 demonstrated a similar trend in gene expression for the CT genotype of 2.0 x10-5 [IQR 0.8 x10-5 - 5.0 x10-5]compared with TT genotype 0.7 x10-5 [0.2 x10-5 – 1.9 x10-5, p =0.080). Analysis for SNP rs712044 demonstrated gene expression for the CT genotype of 1.2x10-5 [IQR 0.7x10-5 - 3.6x10-5]compared with TT genotype 1.1 x10-5 [0.2x10-5 –1.7x10-5, p =0.438).
Correlations between CCL18 mRNA expression and serum CCL18 levels for the rs712040, rs712042 and rs712044 SNP were respectively r=0.46 (p=0.084), r=0.47 (p=0.080) and r=0.09 (p=0.753).
We added to the results that subjects with the CT-genotype of the rs712040, rs712042 and rs712044 SNPs showed similar, but non-significant trends compared with the TT-genotypes. In addition, we added to the text that weak to moderate, non-significant correlations were found between CCL18 mRNA expression and serum CCL18 for the SNPS rs712040, rs712042 and rs712044 (paragraph ‘CCL18 Genotypes and mRNA expression’, lines 193-196)
11. Legend for Fig 4C is too small to differentiate the different groups.
Response from the authors
We enlarged figure 4 in order to clarify the legends for Figure 4.
12. Lines 203-206, is this a three-way analysis? Interaction? Additive model? How was the p-value calculated?
Response from the authors
Survival curves for the three groups (CCL18 low-TT, CCL18 high-TT and CCL18 high-CT) were created using Kaplan Meier curves and compared statistically using the Log Rank Test. We clarified the analysis in the text (paragraph ‘Progression of disease and survival in IPF’, currently lines 229-230) and in the text of Figure 4 (lines 234-237).
13. Line 224 used the abbreviated IIPs without previously defining.
Response from the authors
Thank you for noticing, we defined IIP (idiopathic interstitial pneumonias) in the text (currently lines 249-250).
14. Line 230, clarify the “C genotype”
Respond from the authors
Thank you, we clarified in the text that the results were concerning the CT-genotype of rs2015086 (line 255).
15. It is likely that you see no difference in allele frequency between your cases and controls because all of the controls are too young to have a typical form of IPF. This should be address in the discussion line 241-246
Response from the authors
Thank you for your remark. In general, allele frequencies are not dependent on age. Furthermore, IPF is a rare disease occurring in 494 per 100.000 persons (3). It is therefore unlikely that any- or a significant portion- of the controls will develop IPF in the future. However, we acknowledge that a difference in age exists between the two groups and added this observation to the discussion. We added to the discussion that the difference in age between IPF and healthy controls should be taken into account (currently lines 287-288).
Round 2
Reviewer 1 Report
Satisfactory replies were obtained for questions 1 and 3.
However, the authors have not answered question 2 correctly. Is there no association between the susceptibility to AE and the genotype? Please specify the percentage of patients who develop IPF-AE or the time to first acute exacerbations by genotype of the rs2015086, and explain if there is a difference between the two groups.
Author Response
>Satisfactory replies were obtained for questions 1 and 3.
>However, the authors have not answered question 2 correctly. Is there no association between the susceptibility to AE and the genotype? Please specify the percentage of patients who develop IPF-AE or the time to first acute exacerbations by genotype of the rs2015086, and explain if there is a difference between the two groups.
Original Question from Round 1:
>Acute exacerbation of IPF is usually associated with a poor prognosis. It has been reported that CCL18 production by BAL cells increased during acute exacerbations and baseline levels of CCL18 production by BAL cells were significantly predictive for the development of future acute exacerbation (PLoS One. 2015 Jan 15;10(1):e0116775). Is there difference in the frequency of acute exacerbations or the time to first acute exacerbations depending on genotype?
Adapted Response from the authors
We evaluated progression of disease in IPF patients, based on FVC change and/or the clinical suspicion of an acute exacerbation. An FVC decline of more than 10% after one year was considered as FVC progression. FVC values at one-year follow-up were availabe for 41 out of 77 IPF patients. Median FVC change after one year was -4.5% (IQR -9.4 – 2.3). No significant differences were found in median FVC change (p =0.483) between patients with the CT and TT genotypes of rs2015086.
Of 77 patients, a total of 9 patients (13.2) had a clinical suspicion of an acute exacerbation. No significant difference was found in frequencies of acute exacerbations between the CT-genotype (14.3%) and TT-genotype of rs2015086 (13.0%; p = 0.896).
We added this additional analysis to the results section (paragraph ‘Progression of disease and survival in IPF’, lines 207-208 and 214-216).